# A Bayesian neural ordinary differential equations framework to study the effects of chemical mixtures on survival

Virgile Baudrot[1]*, Nina Cedergreen[2], Thomas Kleiber[1], André Gergs[3], Sandrine Charles[4]

**1** Qonfluens, Montpellier, France, **2** University of Copenhagen, Copenhagen, Denmark, **3** Bayer AG, Monheim, Germany, **4** Université Claude Bernard Lyon 1, Villeurbanne, France

* virgile.baudrot@qonfluens.com

## Abstract

In the field of Plant Protection Products (PPP), the combination of multiple active ingredients is a common strategy to improve the efficacy against target species. However, these mixtures may expose non-target species to untested chemical combinations due to varied application and degradation pathways. Developing a method to predict the effects of such mixtures without extensive testing is crucial for enhancing risk assessment for non-target organisms. Toxicokinetic (TK) and toxicodynamic (TD) models currently provide predictive power for the toxicity of mixture over time, utilizing the concentration addition and independent action principles. However, their capacity to capture complex, non-linear interactions - such as synergistic and antagonistic effects - remains limited. To overcome these limitations, we introduce a new methodology that integrates TKTD models using Ordinary Differential Equations (ODE) with neural networks (NN), offering a robust framework for modeling complex substance interactions. The ODE component encodes fundamental biological principles that constrain the neural network to biologically plausible solutions. Bayesian inference further refines the model by addressing uncertainties in data, model parameters, and biological processes, while also allowing quantification of prediction uncertainty, highlighting gaps in experimental data that warrant further investigation. This hybrid model was evaluated across 99 acute toxicity studies that included various PPP mixtures, testing its ability to identify and forecast deviations from expected mixture behaviors. Our findings demonstrate that while simpler linear models provide a robust and parsimonious baseline for predicting additive mixture effects, the neural network component serves as a powerful tool for selectively identifying and forecasting significant non-linear deviations from these expectations. This dual approach clarifies the distinct roles of both simple and complex models, providing a more responsible framework to anticipate the combined effects of untested chemical mixtures and inform risk assessment practices.

**Data availability statement:** All scripts and data required to replicate the analysis are provided in the Supporting information (S1 Script) and also in the GitHub repository:

https://github.com/virgile-baudrot/maxim_TK_NeuralODEs_TD.

**Funding:** The study was partially funded by Bayer AG (052-2023 Research Agreement between UCBL-Bayer-UCPH-Qonfluens). The funders had no role in study design, data collection and analysis, decision to publish, or preparation of the manuscript.

**Competing interests:** I have read the journal's policy and the authors of this manuscript have the following competing interests: A.G. is employed by Bayer AG, which produces and sells agrochemicals including those investigated in this study. Bayer AG applies TKTD models to assess the effects of chemicals on target and non-target species. The contributions of N.C., V.B., T.K., S.C. to the study were partially funded by Bayer AG (052-2023 Research Agreement between UCBL-Bayer-UCPH-Qonfluens).

## Author summary

Understanding how chemical mixtures affect living organisms is a key challenge in ecotoxicology, as interactions among compounds can lead to additive, synergistic, or antagonistic effects on survival. Traditional toxicokinetic–toxicodynamic (TKTD) models are effective for single compounds but struggle to capture the nonlinear and uncertain dynamics of mixtures. Here, we introduce a Bayesian Neural Ordinary Differential Equations (NODE) framework that merges mechanistic TKTD modeling with deep learning flexibility to describe survival under exposure to chemical mixtures. Our approach was evaluated on four large datasets following OECD Test Guideline 203, including over 500 time-series of fish survival exposed to single and combined pesticides. This model predicts survival across new time points, different mixture ratios, and even unseen mixtures, while quantifying uncertainty through Bayesian inference. The framework performs robustly against established mechanistic models and provides interpretable parameters linked to toxicodynamic processes. By uniting neural differential equations with Bayesian statistics, we offer a new computational paradigm for predicting mixture toxicity—bridging classical mechanistic understanding and modern machine learning, with direct implications for environmental risk assessment and regulatory science.

## 1 Introduction

The use of synthetic pesticides has expanded significantly since its inception in the late 1930s, both in total volume and in the diversity of available active ingredients [7]. Today, more than 500 active ingredients are approved in Europe, distributed in thousands of commercial products [17], resulting in a vast array of chemical mixtures present in ecosystems [33]. These substances may also spread through trophic networks, affecting a wide range of non-target organisms [19]. Active ingredient mixtures arise through multiple pathways: the use of different products in various crops, the combination of products within a single crop, or the incorporation of multiple active ingredients within a single product [33]. While formulated mixtures are generally evaluated for efficacy on target species and regulated for selected non-target species, the effects of unformulated, incidental mixtures in the environment are less well understood. Given the extensive range of substances, concentrations, and combinations involved, it is impossible to empirically test every possible mixture and concentration ratio [5].

For mixtures, the joint effects on exposed organisms are frequently described by two main toxicity models: Concentration Addition (CA) and Independent Action (IA). These models provide reasonable estimates of the effects of the mixture when single-component toxicity data is available for the organism of interest. Reviews indicate that these models accurately predict the effects of joint pesticides in approximately 90% of cases [5,8,28]. However, the remaining cases exhibit interactions that lead to greater or lesser toxicity than predicted, known as synergism and antagonism,

respectively. Such interactions often occur when one chemical (the active substance or the co-formulant) affects the bioavailability, uptake, distribution, or biotransformation of another [8].

Predicting the joint effects of active ingredients is critical for environmental risk assessments, as most plant protection products consist of multiple active ingredients designed to broaden their efficacy and delay resistance. The course of toxic effects over time provides valuable insights for both single substances and chemical mixtures. Toxicokinetic (TK) and toxicodynamic (TD) models are particularly useful here, as they account for the dynamic nature of toxicity. A widely used framework, the General Unified Threshold model of Survival (GUTS), offers a robust method to assess time-dependent mortality trends [23,24], and recent adaptations have extended GUTS to analyze mixture toxicity [2,3,10]. These adaptations aggregate the internal damage caused by each active ingredient, producing a single latent variable to describe the toxicodynamic state of the organism, a concept known as "Damage Addition" [2,3,10,11]. In this approach, the internal damage of each substance is combined additively, providing an input to the TD component of the framework. The underlying assumption is that while substances may have independent TK, their internal effects ultimately converge and accumulate additively within a common TD. It represents an intermediate hypothesis between CA which assumes similar mode of action in both the TK and TD and IA assuming TK and TD pathways can be completely different between substances. While this additive approach of damage allows for predictions across varied exposure profiles, it does not capture non-additive interactions like synergy and antagonism. To refine this approach, some models have adjusted key parameters to represent deviations from additivity. For example, [2] generalizes "Damage Addition" to $n$ active ingredients using a weighted sum of TK outputs, while [10] introduce a deviation parameter that affects rate constants to reflect how one substance may alter the elimination or damage repair rates of another. However, these approaches rely on assumptions that the deviations are likely to stem from interactions with rate-limiting constants, assumptions which remain inadequately tested.

To address these limitations, we propose an alternative approach that minimizes predefined assumptions, utilizing a free-constraint model for damage aggregation between the TK and TD stages by means of Neural Networks (NN). This hybrid model couples the mechanistic TKTD parts of GUTS with a flexible, data-driven NN component to capture complex, non-linear interactions without imposing rigid mechanistic constraints. This "knowledge-embedded" approach, also known as physics-informed neural networks [32] or Neural ODE [12], allows us to retain mechanistic insights for well-understood subcomponents (*e.g.*, absorption processes in TK, mortality mechanisms in TD) while treating more ambiguous interactions as flexible, data-driven functions [21,22]. The NN, constrained by ODE-driven biological laws, is effectively calibrated with large data sets to approximate the complex functional relationships between TK and TD phases.

We designed this hybrid model within a Bayesian framework to incorporate and quantify uncertainties in data, parameters, and biological processes. Bayesian inference, by framing model dependencies as conditional probabilities, allows robust calibration with standard experimental toxicity data (OECD protocols), while also identifying areas where further data collection is needed to refine predictions [25,26]. In this regard, our study has three main objectives: (1) implement a Bayesian-calibrated hybrid model compatible with routine toxicity data; (2) evaluate the predictive capacity of the model in mixtures, including variations in component ratios or entirely new combinations of active ingredients; and (3) assess the model's ability to quantify deviations from CA and IA, characterizing potential synergistic or antagonistic tendencies within these mixtures. To this end, we tested the model using 99 studies on fish survival conducted according to the OECD 203 guidelines [29], covering herbicides, fungicides, and insecticides. Although the parameters of the TKTD model were based on previous work, the NN required extensive testing across multiple configurations, exploring the layer structure, parameter size, and activation functions. This calibration-validation process, with subsets for model calibration and prediction validation, reflects the challenges inherent to data-driven models constrained by smaller effective training sets. Overall, we expect that this hybrid framework advances mixture toxicity modelling by integrating mechanistic insights with the adaptability of neural networks, enhancing the predictive accuracy of untested chemical combinations and supporting more reliable environmental risk assessment.

## 2 Methods

### 2.1 Data sets

The inference process was tested on 99 acute toxicity studies (data in the S1) with rainbow trout (according to the OECD 203 standards) [29], divided into four data sets comprising 38, 19, 21, and 21 studies, respectively. The experiments consisted of 96 hours of exposure, in which survival was monitored daily over a range of 4 to 8 concentrations tested, constant over time. For most of the data sets, the concentration range allowed one to cover low to high mortality. Each data set facilitated the analysis of interactions among 5 to 6 different active ingredients.

The data set 1 consists of 244 time-series covering 21 mixtures, resulting in 1482 survival data points. The active ingredients are: bixafen, fluopyram, prothioconazole, spiroxamine, tebuconazole and trifloxystrobin. For instance, in this large data set, there are 13 times series with only bixafen as active ingredients, 6 time-series of a mixture of bixafen and fluopyram, 6 other times-series with the three active ingredients, bixafen, fluopyram, prothioconazole, another mixture with prothioconazole and spiroxamine (and others as detailed in the Supporting Information Tables A to H in S1 Appendix). In the same way, the data set 2 consists of 120 time-series covering 9 mixtures, resulting in 662 survival data points. The active ingredients are deltamethrin, flupyradifurone, spiromesifen, tralomethrin and triazophos. The data set 3 consists of 130 time-series covering 11 mixtures, given 727 survival data points. The active ingredients are beta-cyfluthrin, clothianidin, cyfluthrin, imidacloprid, thiacloprid and thiodicarb. Finally, the data set 4 consists of 126 time-series covering 12 mixtures, given 758 survival data points. The active ingredients are aclonifen, diflufenican, flufenacet, flurtamone and metribuzin.

Following guidelines, the survival endpoint refers to no visible movements such as gill movements and no reaction when touching of the caudal peduncle, while observation of immobility was considered as abnormal swimming behaviour. To further evaluate how model calibrations under the Concentration Addition hypothesis would behave in scenarios of synergism or antagonism compared to purely additive interactions, we generated three artificial data sets, as illustrated in Fig 1.

#### 2.1.1 Potential effect of co-formulant.
To ensure that co-formulants do not affect survival, we compared the LC50 predicted with the formulation to the LC50 predicted with the active ingredients alone. This comparison was conducted using single-compound formulations that share the same co-formulants as those used in formulations containing mixtures of several active ingredients. No additional effects of the co-formulants were detected (see Sect 3 in S1 Appendix).

### 2.2 Model

The model is organized into three submodels, as illustrated in Fig 2. The first and third submodels correspond to the toxicokinetic (TK) and toxicodynamic (TD) components, respectively. The TK sub-model describes the dynamics of the internal chemical concentration within the organisms, while the TD sub-model translates this internal concentration into biological damage and calculates the survival probability of the organisms. Both sub-models utilize mixtures from the reduced General Unified Threshold model of Survival (GUTS) [24]. The second intermediary submodel bridges TK and TD by translating the TK output (internal chemical concentrations) into a single input for the TD model (biological damage) using several aggregation mixtures. Throughout this paper, we use the term "aggregation" of substances to describe this intermediary process, distinguishing it from "interaction" which refers to the biochemical interactions between substances.

#### 2.2.1 Toxicokinetics.
The toxicokinetic (TK) component of the model is mechanistic, defined by Ordinary Differential Equations (ODE) to capture the time-dependent dynamics of the exposure profiles. For each substance $i$ with an exposure profile $X_i(t)$ in water, the TK model describes the kinetics of the substance as a proxy for the scaled internal concentration, denoted $C_i(t)$, since direct measurements of internal concentrations are not available. This model includes the dominant toxicokinetic rate constant, $k_{d,i}$, which is specific to each substance $i$. The TK model is represented by the following equation:

$$\frac{dC_i(t)}{dt} = k_{d,i}\left(X_i(t) - C_i(t)\right) \tag{1}$$

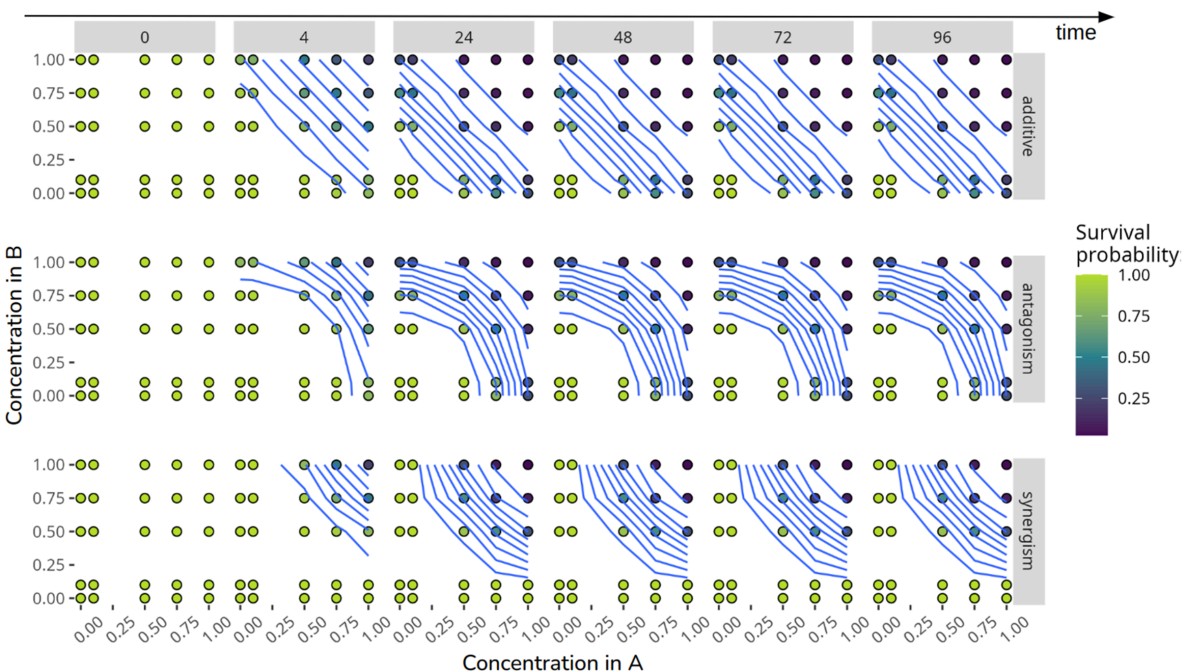

**Fig 1**. **Artificial data sets were generated for a mixture containing two active ingredients, A and B, with interaction patterns represented by blue effect isoclines, indicating equal survival probabilities over time (0, 4, 24, 48, 72, and 96 hours, displayed across columns).** The concentration and constituent proportions of the mixture are illustrated by dots. When compared to Concentration Addition hypothesis, the top row displays the additive interaction pattern, the middle row shows antagonism, and the bottom row depicts synergism.

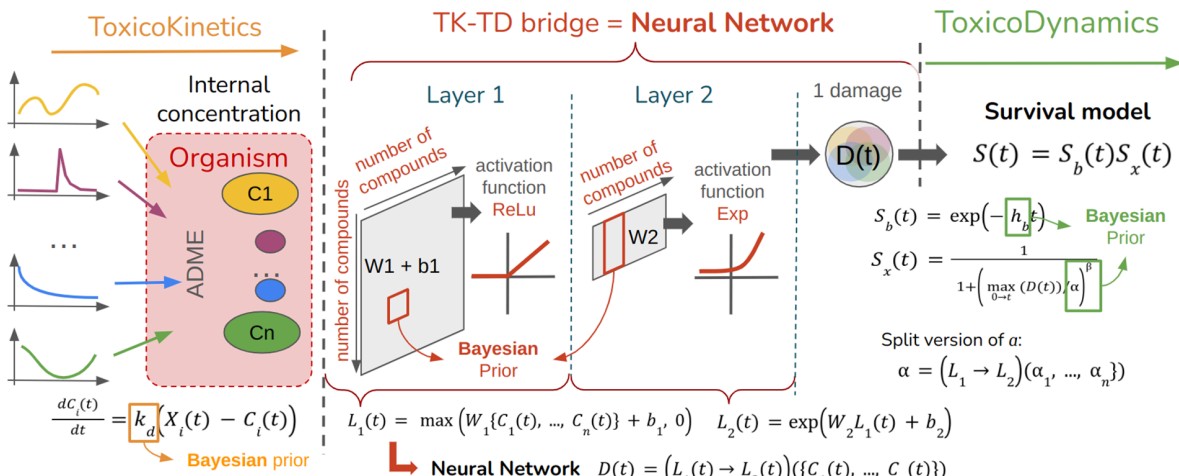

**Fig 2**. **Diagram of the generic models illustrating the sequential steps, beginning with (1) the toxicokinetic modeling, which describes the uptake and elimination of *n* compounds; followed by (2) the bridging component between the toxicokinetic and toxicodynamic sections, which aggregates the compounds into a one-dimensional, time-dependent damage variable, *D(t)*; and concluding with (3) the toxicodynamic modeling that translates *D(t)* into survival probability.**

**2.2.2 TKTD bridge.** The second part of the model serves as the bridge between the TK and TD components, with multiple variants detailed in Table 1.

The bridging function operates on the scaled internal concentration of each substance, $C_i(t)$. To calculate the damage, there are two possible approaches: using either the derivative of $C_i(t)$, or the variable itself. For $n$ exposure compounds, using the derivative approach yields the following:

$$\frac{dD(t)}{dt} = f_d\left(\frac{dC_1(t)}{dt}, \dots, \frac{dC_n(t)}{dt}\right) \tag{2}$$

where $f_d()$ is a function of the derivatives for each substance $i$.

Alternatively, working directly with the concentrations gives:

$$D(t) = f(C_1(t), \dots, C_n(t)) \tag{3}$$

where $f()$ is a function of all concentrations of substances.

We chose the second option as a straightforward rule that ensures that the variable $D(t)$ remains positive, which we achieved by applying an exponential transformation directly to $D(t)$.

**Linear model.** If the model is linear, denoted `n` in Table 1, the equations are similar to those used in previous studies [2,3,10], with the addition of a bias $b$ to obtain a true linear model.

**Neural network.** To address non-linear aggregations, we implemented a Neural Network (NN) based on the Universal Approximation Theorem, which states that a two-layer NN with a non-linear activation function can approximate any function [13].

The input for the TKTD bridge is the scaled internal concentration vector of each substance $i$, represented as $\mathbf{C}(t) = (C_1(t), \dots, C_i(t), \dots C_n(t))$, while the output is the calculated damage, $D(t)$.

**Table 1**. Labels and mathematical equations for the TKTD bridge models used in calibration and depicted in Fig 3.

| Labels | Equations |
|---|---|
| n | $D(t) = b + \sum_i a_i C_i(t)$ |
| n_exp | $D(t) = \exp\left(b + \sum_i a_i C_i(t)\right)$ |
| nn_n_exp | $\mathbf{z}_1(t) = \mathbf{W}_1 C(t) + \mathbf{b}_1$ <br> $D(t) = \exp\left(\sum_i a_i z_{1,i}(t)\right)$ |
| nn_ReLU_n_exp | $\mathbf{z}_1(t) = \max\left(\mathbf{W}_1 C(t) + \mathbf{b}_1, 0\right)$ <br> $D(t) = \exp\left(\sum_i a_i z_{1,i}(t)\right)$ |
| nn_ReLU_nn_ReLU_n_exp | $\mathbf{z}_1(t) = \max\left(\mathbf{W}_1 C(t) + \mathbf{b}_1, 0\right)$ <br> $\mathbf{z}_2(t) = \max\left(\mathbf{W}_2 z_1(t) + \mathbf{b}_2, 0\right)$ <br> $D(t) = \exp\left(\sum_i a_i z_{2,i}(t)\right)$ |
| nn_ReLU_nn_ReLU_nn_ReLU_n_exp | $\mathbf{z}_1(t) = \max\left(\mathbf{W}_1 C(t) + \mathbf{b}_1, 0\right)$ <br> $\mathbf{z}_2(t) = \max\left(\mathbf{W}_2 z_1(t) + \mathbf{b}_2, 0\right)$ <br> $\mathbf{z}_3(t) = \max\left(\mathbf{W}_3 z_2(t) + \mathbf{b}_3, 0\right)$ <br> $D(t) = \exp\left(\sum_i a_i z_{3,i}(t)\right)$ |

The first layer performs a linear aggregation of input features, adding a bias term:

$$\mathbf{z_1}(t) = \mathbf{W_1}\mathbf{C}(t) + \mathbf{b_1} \tag{4}$$

where $\mathbf{W_1} = \{w_{i,j}\}_{i,j=1,n}$ is an $n \times n$ weight matrix, and $\mathbf{b_1} = \{b_{1,i}\}_{i=1,n}$ the bias vector of size $n$. This yields $\mathbf{z_1}(t)$, a vector of size $n$ with each coordinate given by: $z_{1,i}(t) = \sum_{j=1}^{n} w_{i,j}C_j(t) + b_{1,i}$.

At this stage, the model remains linear. To introduce non-linearity, we apply an activation function, $\sigma_1$, to the resulting $\mathbf{z_1}(t)$:

$$\mathbf{a_1}(t) = \sigma_1(\mathbf{z_1}(t)) \tag{5}$$

In the final model configuration, we used a rectified linear unit function (ReLU) as the activation function, defined as $\sigma_1(\mathbf{z_1}(t)) = \max(0, \mathbf{z_1}(t))$. The combination of the linear operation in (4) with the activation function, (5), creates a perceptron defining the first layer of the neural network as $L_1(\mathbf{C}(t)) = \sigma_1(\mathbf{W_1}\mathbf{C}(t) + \mathbf{b_1})$. For the second layer, a similar approach is applied:

$$\mathbf{a_2}(t) = \sigma_2(\mathbf{z_2}(t)) \text{ with } \mathbf{z_2}(t) = \mathbf{W_2}\mathbf{z_1}(t) + \mathbf{b_2} \tag{6}$$

All models used for calibration are presented in Table 1.

**2.2.3 Toxicodynamics.** The component of Toxicodynamics (TD) is the final stage of the model, receiving as input the latent damage variable $D(t)$, derived from the combined output variables of the component of Toxicokinetics (TK) through the TKTD bridge (see Sect 2.2.2). This aggregate damage, $D(t)$, represents the cumulative marginal damage $D_i(t)$ for each active ingredient $i$.

The Individual Tolerance version of the General Unified Threshold model of Survival (GUTS-IT) framework is then employed, incorporating both the baseline survival probability $S_b(t)$ and the additional survival probability due to exposure to each substance $i$. The role of the TD component is only to convert cumulative damage $D(t)$ into a global survival probability $S(t)$, expressed as follows:

$$S(t) = \exp(-h_b t)\left(1 - \frac{1}{1 + \left(\frac{\max\limits_{0 \le \tau \le t}(D(\tau))}{\alpha}\right)^{-\beta}}\right) \tag{7}$$

This equation satisfies the initial conditions for damage, specifically that the damage is zero at the beginning of the experiment, $D(t=0) = 0$, and remains positive for all time points $t$, $D(t) \ge 0\, \forall t$). In its simplest form, the TD component relies on three parameters: background mortality, $h_b$, median $\alpha$ and shape $\beta$ of the threshold distribution under the GUTS-IT framework.

**Splitting $\alpha$.** In a previous study, we demonstrated that the parameter $\alpha$ is correlated with the parameter that represents the no-effect concentration [4]. To account for such a correlation within the context of mixtures, we propose an alternative model, termed "splitting", in which $\alpha$ is expressed as a function of the marginal $\alpha_i$ for each individual compound $i$. This approach utilizes the same function used to model damage from scaled internal concentrations, as follows:

$$\alpha = f(\alpha_1(t), ..., \alpha_n(t)) \tag{8}$$

where $f()$ is the same function as the one defined in Eq (3).

### 2.3 Interaction hypotheses

**2.3.1 Concentration Addition (CA).** Let us consider two compounds $A$ and $B$, with concentrations $C_A$ and $C_B$. The concept of Concentration Addition (CA) states that the combined effect of the two compounds can be predicted by adding their concentrations in proportion to their isolated effect concentrations. Mathematically writing, denoting $s()$ the survival probability, then the lethal concentrations that lead to $X\%$ of effects, $LC_{X,A}$ and $LC_{X,B}$, are defined as $s(LC_{X,A}) = s(LC_{X,B}) = 1 - X/100$. The CA hypothesis is satisfied if for any proportion $p$: $s(p\,LC_{X,A} + (1-p)\,LC_{X,B}) = s(LC_{X,A}) = s(LC_{X,B})$.

Therefore, assuming $s(LC_{X,A}) = s(LC_{X,B}) = s(C_A, C_B)$, then CA explains the mixture if $\frac{C_A}{LC_{X,A}} + \frac{C_B}{LC_{X,B}} = 1$. This approach has been generalized by [6] and [18] to provide a generalized CA hypothesis with $n$ compounds and to measure the deviation from it:

$$\sum_{i=1}^{n} \frac{C_i}{LC_{X,i}} = 1 \tag{9}$$

In the situation where survival under exposure to the mixture is lower than predicted for compound A alone, $s(p\,LC_{X,A} + (1-p)\,LC_{X,B}) < s(A)$. This effect is termed "CA synergism", where the observed effect exceeds what would be expected by CA. If survival is higher than predicted, this is referred to as "CA antagonism", and is expressed as $s(p, LC_{X,A}\,(1-p), LC_{X,B}) > s(A)$.

The "Damage Addition" approach, as applied in previous studies [2,3,10] (model `n` in Table 1), is based on the sum of the internal concentrations estimated by the TK component, formulated as $D(t) = \sum_{i=1}^{n} C_i(t)$. This aggregate value is then scaled according to the effect of each chemical using the values $LC_X$, as described above. Using the simplified TK model of the reduced GUTS, where there is only one parameter per substance, $k_{d,i}$, relating internal to external concentration, a single $k_d$ value can be found, allowing multiple substances to share the same TK characteristics. Thus, in our model, "Damage Addition" aligns directly with CA, along with all of its underlying assumptions.

**2.3.2 Independent Action (IA).** The IA hypothesis is expressed by the equation $s(C_A + C_B) = s(C_A) \times s(C_B)$, which is often represented in terms of effect $e$ as: $e(C_A + C_B) = 1 - (1 - e(C_A)) * (1 - e(C_B))$ [18]. Accordingly, deviation from the IA hypothesis is defined differently than for CA. Specifically, there is "IA synergism" if $s(C_A + C_B) < s(C_A) \times s(C_B)$ and "IA antagonism" if $s(C_A + C_B) > s(C_A) \times s(C_B)$. [18] provided a general definition for the deviation in effect when considering $n$ compounds as follows:

$$e_{mixture} = 1 - \prod_{i=1}^{n}(1 - e(C_i)) \tag{10}$$

**2.3.3 Deviation.** To quantify deviations from the CA and IA hypotheses, we adopted a deviation equation used in previous studies [5,10,18], where a ratio is defined between the expected outcomes under each hypothesis and the observed outcomes. To facilitate comparisons across multiple mixtures at varying time points and effect levels, we defined a symmetric and normalized deviation ratio, called Mean Penalized Deviation (MPD), expressed as follows:

$$\mathbf{MPD} = \frac{1}{n}\sum_{i=1}^{n}\left(\frac{\text{benchmark}_i - \text{predicted}_i}{\text{benchmark}_i + \text{predicted}_i} \times (1 - \text{uncertainty}_i)\right) \tag{11}$$

The term predicted$_i$ refers to the value calculated by our model, while benchmark$_i$ corresponds to the expected value under the CA or IA hypotheses. Given that both benchmark$_i$ and predicted$_i$ are positive, the resulting Mean Penalized Deviation (MPD), falls within the interval $[-1,1]$. A deviation of 0 indicates the match to the CA or IA hypothesis; a negative deviation indicates a tendency toward antagonism and a positive deviation suggests synergism. The term uncertainty$_i$ represents the uncertainty of predicted$_i$, estimated by our model within a Bayesian framework and captured by the range between the 97.5 and 2.5% quantiles of the posterior distribution.

## 2.4 Implementation

Simulation and inference were performed using R with the combination of the libraries `deSolve` R-package for simulation [34] and the `JAGS` software [30] via the `rjags` R-package for Bayesian inference [31]. All scripts and data are available in the Supporting Information S1 Script (and also at Github repository: https://github.com/virgile-baudrot/maxim_TK_NeuralODEs_TD).

During the calibration phase, we fitted the four data sets using eight different models, each sharing the same TK and TD components but differing in their specific TKTD bridge expressions, as described in Table 1. The simplest mode model, denoted n in Table 1, is linear. From this baseline, the complexity of the model was incrementally increased by adding one or more layers, incorporating weight matrices, biases, and activation functions.

For the MCMC configuration, we ran three chains, each with 7500 iterations, (5000 for warm-up and 2500 for chain convergence). We checked the convergence of the MCMC using the Gelman-Rubin diagnostic (R<1.05 for all parameters), and ensuring an Effective Sample Size (ESS) greater than 400 to guarantee a stable estimation of the posterior distribution. We also checked the traces to confirm good mixing and the absence of trends. The priors distribution are given in the Table 2.

## 2.5 Goodness-of-fit

To assess the quality of the calibration output, we used the model performance criteria recommended by EFSA in its Scientific Opinion on TKTD modeling [15], including the Posterior Prediction Check (PPC), the Normalized Root Mean Square Error (NRMSE), and the Survival Probability Prediction Error (SPPE).

The PPC enables a comparison between the predicted and observed survival counts, summarizing the predictions by their mean and the 95% uncertainty range. The percentage of predictions that capture the observed data serves as a quantitative goodness-of-fit criterion.

The NRMSE is defined by the equation:

$$\text{NRMSE} = \frac{1}{\frac{1}{N}\sum_i y_{obs,i}} \sqrt{\frac{1}{N}\sum_i \left(y_{obs,i} - y_{pred,i}\right)^2} \tag{12}$$

where $y_{obs,i}$ and $y_{pred,i}$, $i = 1, N$, are the observed and predicted values, respectively, across $N$ data points.

The SPPE focuses on the survival probabilities at the end of the experiment, quantifying the accuracy of the model at the end of the exposure period. It is defined as:

$$\text{SPPE} = \frac{y_{obs,t_{end}} - y_{pred,t_{end}}}{y_{obs,t_0}} \tag{13}$$

where $y_{obs,t_{end}}$ and $y_{pred,t_{end}}$ represent the observed and predicted survival counts at the end of the exposure period.

**Table 2. Prior distributions for all model parameters.** For the parameter column, we specified the model process in square brackets: [TK] for the toxicokinetic model part, [NN] fot the TKTD bridge with the Neural Network, and [TD] for the toxidynamic part.

| Parameter | Description | Distribution | Hyperparameters |
|---|---|---|---|
| $k_{d,i}$ [TK] | Dominant TK rate for substance $i$ | log-Normal | Based on single-substance analyses |
| $W_k$ [NN] | Weight matrix for layer $k$ | Normal | Mean=0, Standard Deviation=0.5 |
| $b_k$ [NN] | Bias vector for layer $k$ | Normal | Mean=0, Standard Deviation=0.5 |
| $h_b$ [TD] | background mortality rate | log-Normal | Derived from the control group data of each experiment |
| $\alpha$ [TD] | Median of threshold distribution | log-Normal | Based on single-substance analyses |
| $\beta$ [TD] | Shape of threshold distribution | log-Uniform | Range in $[10^{-2}, 10^2]$ |

Since Bayesian inference was applied, we included the Widely Applicable Information Criterion (WAIC) to estimate the effective number of parameters, accounting for overfitting [36]. The WAIC is defined as:

$$\text{WAIC} = -2\,(\text{lpd} - \text{pWAIC}) = -2\left(\sum_i \log(\mathbb{E}(p(y|\theta))) - \sum_i \text{var}(\log(p(y|\theta)))\right) \tag{14}$$

where lpd is the log-point-wise predictive density and pWAIC represents the effective number of parameters.

It is important to note that both NRMSE and WAIC are relative metrics. Their absolute values are not meaningful in isolation; instead, they are used to compare the performance of different models that have been calibrated on the same dataset. For NRMSE, a model with an NRMSE of 0.02 has a root mean square error twice as large as a model with an NRMSE of 0.01, indicating a comparatively poorer fit to the data. Regarding WAIC, estimating predictive accuracy while penalizing for complexity, it is commonly accepted that a difference between models of less than 2 is considered negligible, while a difference of more than 10 suggests a strong error.

## 2.6 Cross-validation

The cross-validation phase involved dividing each data set into two subsets: one for calibration, known as the training data set, used for parameter inference, and the other for validation, where observed data were compared with model predictions post-training. We implemented two approaches for splitting the data sets into training and validation subsets. In the *random time-point* approach, 90% of the data points were chosen randomly for calibration, while the remaining 10% were reserved for validation. In the *random time-series* approach, selection occurred at the level of entire time series, maintaining the same 90/10% ratio for training and validation (see Tables A and B in S1 Appendix). We could have designed the training and validation subsets in a more structured way, for example to specifically test predictions on reformulation or predictions at new time points. However, since time and concentration are dynamically coupled through the ODE-based toxicokinetic model, a new concentration ratio inherently generates a new temporal trajectory. As a result, these two types of predictions cannot be evaluated independently.

## 2.7 Validation on independent mixtures

We conducted an additional validation phase using independent mixtures, specifically data sets that were not included in the calibration phase. To achieve this, we removed all time series corresponding to mixtures containing pairs of active ingredients prior to the calibration phase. The model was then fitted to the remaining training data set, and the predictions were subsequently compared with the removed time series during the validation phase (see Fig D in S1 Appendix). For illustration, consider a data set with three active ingredients: A, B, and C. The complete data set would likely include survival time series for various mixtures with one, two, or three active ingredients: *A,B,C,AB,AC,BC,ABC*. If we designate the pair *AB* for removal from the training subset, leaving *A,B,C,AC,BC,ABC*, then the validation phase would involve comparing the model predictions with the observed survival under the *AB* exposure profiles.

## 2.8 Prediction

Unlike the validation phase, the prediction phase lacks observational data to assess the prediction quality. There are three types of prediction (see Fig E in S1 Appendix): (1) predictions for the same mixture as in calibration but at new time points; (2) predictions for the same mixture as in calibration but with different exposure ratios; and (3) predictions for newly composed mixtures of tested active ingredients not yet tested.

## 3 Results

### 3.1 Calibration

Model calibration was performed separately for the three artificial data sets and the four empirical data sets. The Fig 3 provides an overview of the quality of model fit based on two goodness-of-fit measures: the Widely Applicable Information Criterion (WAIC) and the Normalized Root Mean Square Error (NRMSE). Models with dots located in the bottom left corner of each graph indicate a good fit.

For artificial data sets (upper panel in Fig 3, the linear models `n` and `n_exp` best fit the data simulated with the additive model. In contrast, these two models perform poorly when calibrated on antagonistic and synergistic data, displaying high NRMSE and WAIC values. The most complex model, with three NN layers (`nn_ReLU_nn_ReLU_nn_ReLU_n_exp`), is the least effective for the additive data but performs best for the antagonistic and synergistic data (Fig 3).

Although NRMSE quantifies the fit between observations and predictions, WAIC accounts for the cost associated with the number of model parameters. As shown in Fig 3, calibration results for artificial data reveal that the high parameter demand and computation time of the NN models is effectively offset when non-linearities are involved in compound aggregation. In simple terms, although NN models are complex and time-intensive, their application proves highly advantageous and thus warranted when handling mixture data with non-linear aggregations (e.g., involving antagonism and

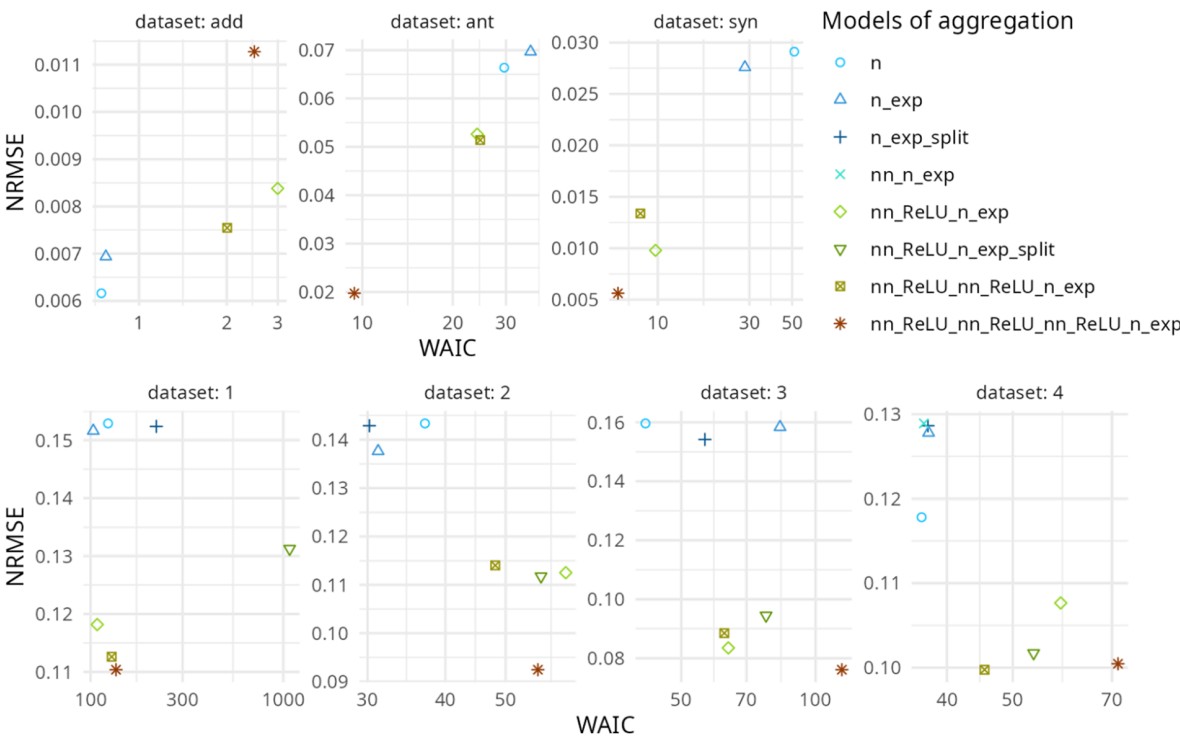

**Fig 3**. **Scatter plots illustrate the quality of calibration results for three artificial data sets fitted to five generic models (upper panel) and for four experimental data sets fitted to eight different models (lower panel). The WAIC ($x$-axis) and NRMSE ($y$-axis) serve as goodness-of-fit metrics, with lower values indicating higher fit quality**. In the "Models of aggregations" legend, `n` stands for the simplest linear model with $n$ compounds ($b + \sum_{i=1}^{n} a_i C_i$); `n_exp` for the linear model with an exponential transformation ($\exp(b + \sum_{i=1}^{n} a_i C_i)$); `n_exp_split` is the `n_exp` model with the splitting of $\alpha$; `nn_n_exp` is the application of a perceptron matrix of size $n \times n$ without activation function before the `n_exp` model; and `nn_ReLU_n_exp` is the application of an $n \times n$ layer with a `ReLU` function before the `n_exp` model. The remaining models with the `nn_ReLU` prefix involve a sequence of perceptron layers before the `n_exp` model. See Table 1 for details. The data sets called "add", "ant" and "syn" were simulated under additive, antagonism and synergism interaction hypotheses, respectively. See Fig 1 for details.

synergism). In contrast, when the concentration addition (CA) hypothesis is met, using an NN model for damage aggregation becomes excessive, and a simple linear model suffices.

Calibration on empirical data also yields NRMSE values, indicating that NN models consistently outperform linear models. When including a cost adjustment for model parameters (as in WAIC), linear models (blue dots in Fig 3) show slightly better performance for data sets 2 and 4, while performance is comparable for the others.

To determine whether the empirical data satisfy the CA hypothesis, we compared the calibration performance in Fig 4 with the results of the artificial data (Fig 3). This comparison suggests the presence of non-linearities in each of the four empirical data, as NN models consistently perform better. If the CA hypothesis were largely valid, models with linear aggregation (without NN) would have demonstrated the best performance.

### 3.2 Validation

The cross-validation results (Fig 4) indicate that the *random time-series* validation approach yields significantly better outcomes compared to the *random time-points* approach (see Tables A and B in S1 Appendix). For both cross-validation approaches and across all four experimental data sets, we applied the top four models to perform cross-validation, with each approach repeated five times. Using NRMSE, the variability across these five repetitions (Fig 4) demonstrates that no model consistently outperforms the others, regardless of the data set. For the approach using time-points, in mean (standard deviation) the NRMSE are respectively for no-split and split linear models of 0.877 (0.0887), 0.912 (0.0816) models and for no-split and split NN models of 0.918 (0.0857), 0.876 (0.0813) ; and for the approach on the time-series, in mean the NRMSE are respectively of 0.400 (0.0814), 0.409 (0.0856) for linear models and 0.419 (0.0905), 0.422 (0.0907) for models with NN.

For the validation for completely unknown mixtures, the results shown in Fig 5 indicate that models without NN components (blue points) are the most effective, except in the cases of the flupyradifurone-spiromesifen pair in data set 2 and the diflufenican-metribuzin pair in data set 4. Summary statistics from Table 3 show that, on average (both in terms of mean and median), the linear model performs better than models that include a Neural Network.

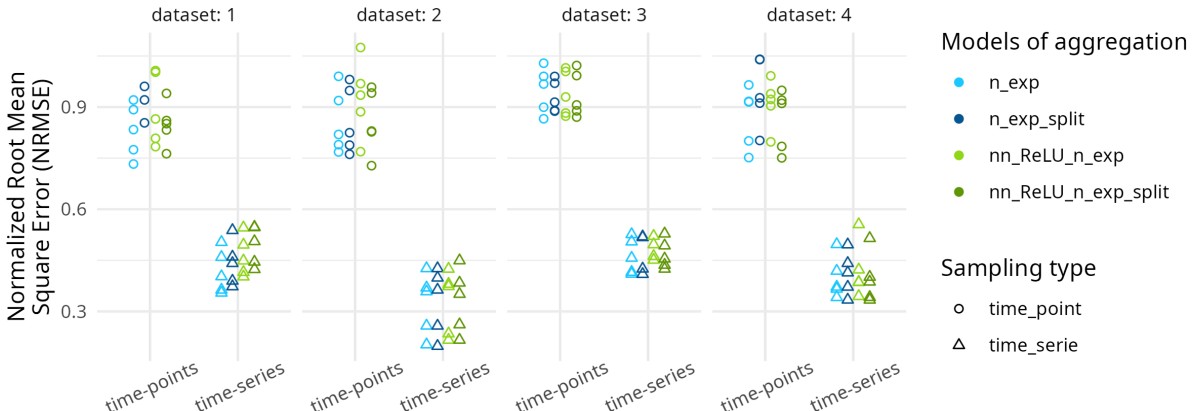

**Fig 4**. **Dots represent NRMSE values obtained by fitting the four models (distinguished by different colors) across the four experimental data sets: lower NRMSE values indicate higher-quality cross-validation results**. The x-axis is not quantitative (points are spread). Two sampling approaches are shown: random selection of time-points (circle dots) and random selection of time-series (triangle dots), each with 90% of the data set used for calibration (training) and 10% for validation. Each cross-validation was repeated five times, illustrated by five identical points.

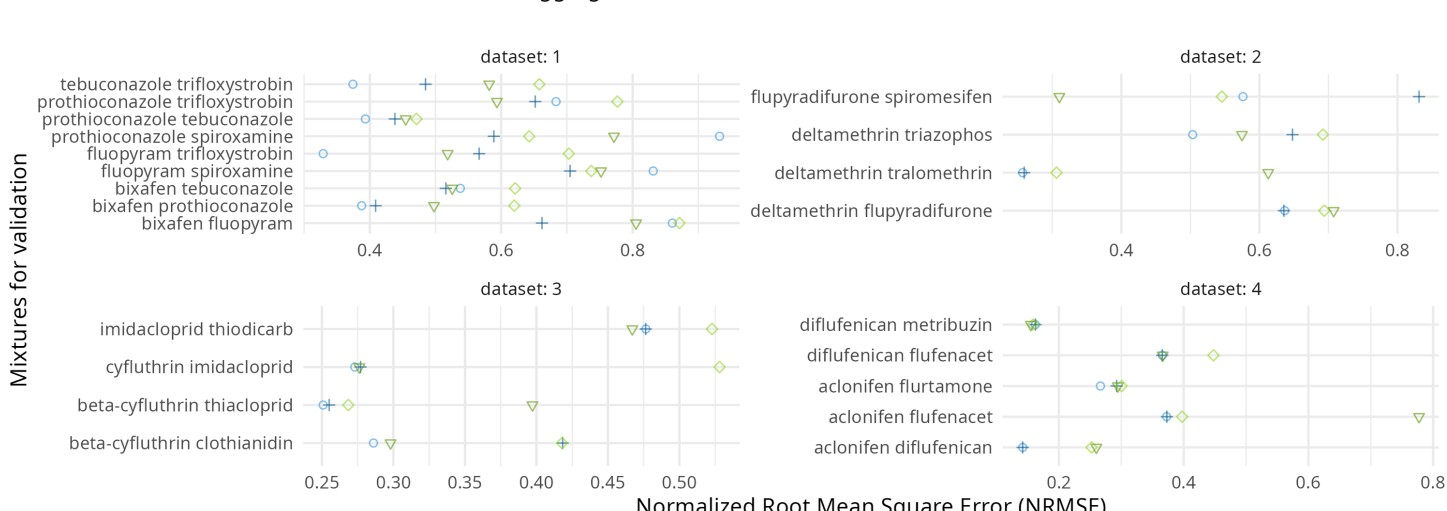

**Fig 5**. The x-axis represent the NRMSE values obtained for the four models (distinguished by different colors) when predicting entirely new mixtures composed of combinations of active ingredients not present in the experimental data sets.

**Table 3**. Global statistics of the linear and NN models used for validation on new mixture summarizing outputs seen in Fig 5.

| Models | NRMSE mean (sd) | NRMSE median (95%CI) |
|---|---|---|
| linear (n_exp) | 0.450 (0.224) | 0.381 (0.567, 0.895) |
| linear (n_exp_split) | 0.462 (0.186) | 0.457 (0.624, 0.765) |
| NN (nn_ReLU_n_exp) | 0.529 (0.192) | 0.537 (0.684, 0.822) |
| NN (nn_ReLU_n_exp_split) | 0.500 (0.191) | 0.508 (0.608, 0.791) |

### 3.3 Prediction

The predictions use parameter values and their probability distributions inferred during the calibration phase performed under a Bayesian framework. These distributions are then used as inputs into the model to predict survival at various times, mixture proportions, and for new mixtures.

An effect prediction and the associated uncertainties for different time points and a mixture of varying proportions of two chemicals are shown in Fig 6 for the example of deltamethrin and tralomethrin. Uncertainty is quantified by the extent of credible intervals resulting from the propagated uncertainties of parameter estimates obtained through Bayesian inference. We used the 95% credible intervals (CI) of the predicted survival rate as an uncertainty measure. The width of this interval serves directly as an uncertainty measure because the survival interval is bounded in [0,1].

In Fig 6 (middle and lower panels), white indicates low uncertainty (95% CI near 0), while purple represents high uncertainty (95% CI near 1). As expected, the uncertainty heat map at time 0 is entirely white. For later time points, white areas surround the points or occupy the upper right corners of heat maps where the survival probability drops to 0. The purple regions, where uncertainty peaks, indicate critical scenarios in which minor adjustments in the relative proportions of the compounds lead to substantial shifts in survival probabilities. This observation is particularly valuable, as it demonstrates that the uncertainty interval (95% CI here) serves as an effective metric for the quality of the prediction.

In addition, complex models may occasionally produce predictions disconnected from the input data. This is especially relevant for hybrid models combining mechanistic components and neural networks. Due to the constraints imposed by

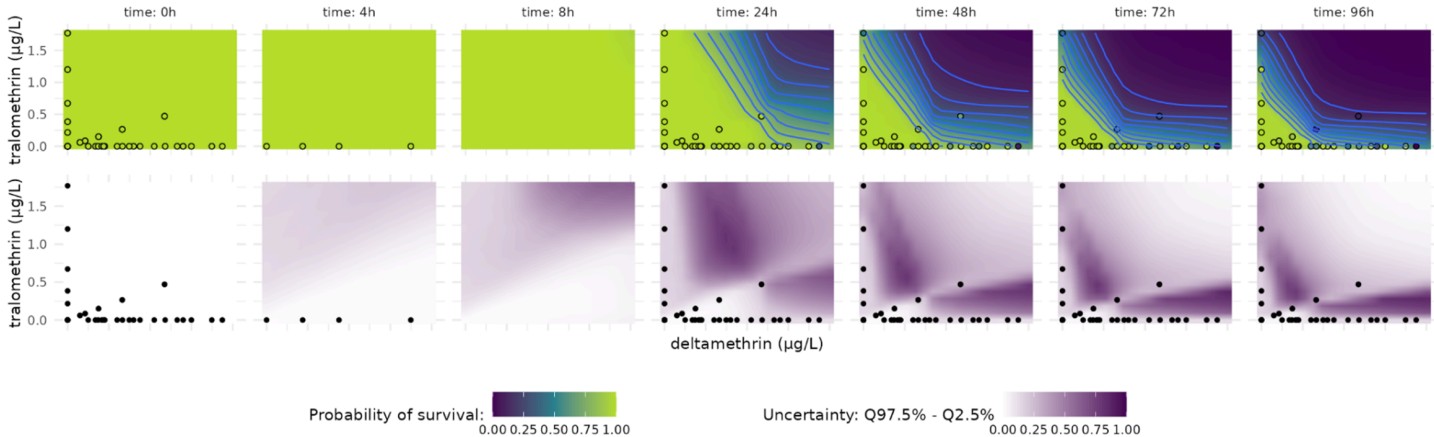

**Fig 6**. **In the top panel, heat maps display the predicted survival probabilities for a binary mixture across different time points, using the neural network model `nn_ReLU_n_exp`.** Light blue lines indicate survival probability isoclines, where the shapes reflect patterns of interaction. In the lower panel, heat maps illustrate the uncertainty of the predictions, calculated as the range of their 95% credible intervals.

the mechanistic model component, outputs are generally kept within a plausible range, making errors (sometimes termed "hallucinations") harder to identify. This uncertainty measure enables the identification of such "hallucination" regions, or at least regions where further experimental data would improve predictions by reducing uncertainty.

**3.3.1 Deviation in prediction.** The final results of this study, illustrated in Fig 7, focus on the deviation from the CA and IA hypotheses. For each prediction (Fig 6), we calculated, for each data point, a deviation relative to the expected results under the CA or IA hypotheses. We then calculated the mean deviation for all points, weighted by an uncertainty penalty (see Eq (11)). To mitigate the influence of high uncertainty, we excluded regions where uncertainty exceeded 50% (i.e., where the 95% credible interval spanned more than 50%). This step was necessary because we summarize each prediction using a single "mean penalized deviation" value, and including points with excessively large uncertainty could distort the interpretation of the overall results. Removing a point indicates that the model failed to provide a reliable prediction.

Data points indicate the deviation calculations for all four aggregation models, and each row on the graphs represents one of the chemical pair tested. A value of zero on the *x*-axis indicates no deviation, signifying that the CA or IA hypothesis holds. In contrast, negative or positive values denote a potential for antagonistic or synergistic effects, respectively. This deviation metric ranges from -1 to 1, where 0 denotes no deviation, and -1 or 1 are the highest (asymptotically) deviations. For effect quantification, we selected thresholds of 0.1 for a weakly probable effect and 0.2 for a more certain effect. Across all predictions, there is no evidence of antagonistic effects and only a few instances of synergistic effects. The CA and IA hypotheses hold in general across all pairs. Two pairs, clothianidin-thiodicarb and clothianidin-beta-cyfluthrin, deviate from the CA hypothesis, suggesting a potential for synergistic effect (Fig 7). For the IA hypothesis, synergistic effects were observed for the pairs trifloxystrobin-bixafen, flurtamone-aclonifen, and deltamethrin-tralomethrin, but with a large uncertainty in the prediction for all three couples.

## 4 Discussion

This study pursued three primary objectives by developing and applying a hybrid model that integrates mechanistic toxicokinetic-toxicodynamic (TKTD) components with neural network (NN) elements, implemented within a Bayesian framework. Our first objective was to demonstrate the practical applicability of this model with standard toxicology data sets. We achieved this by applying it to 99 data sets on fish survival exposed to various types of pesticides, including

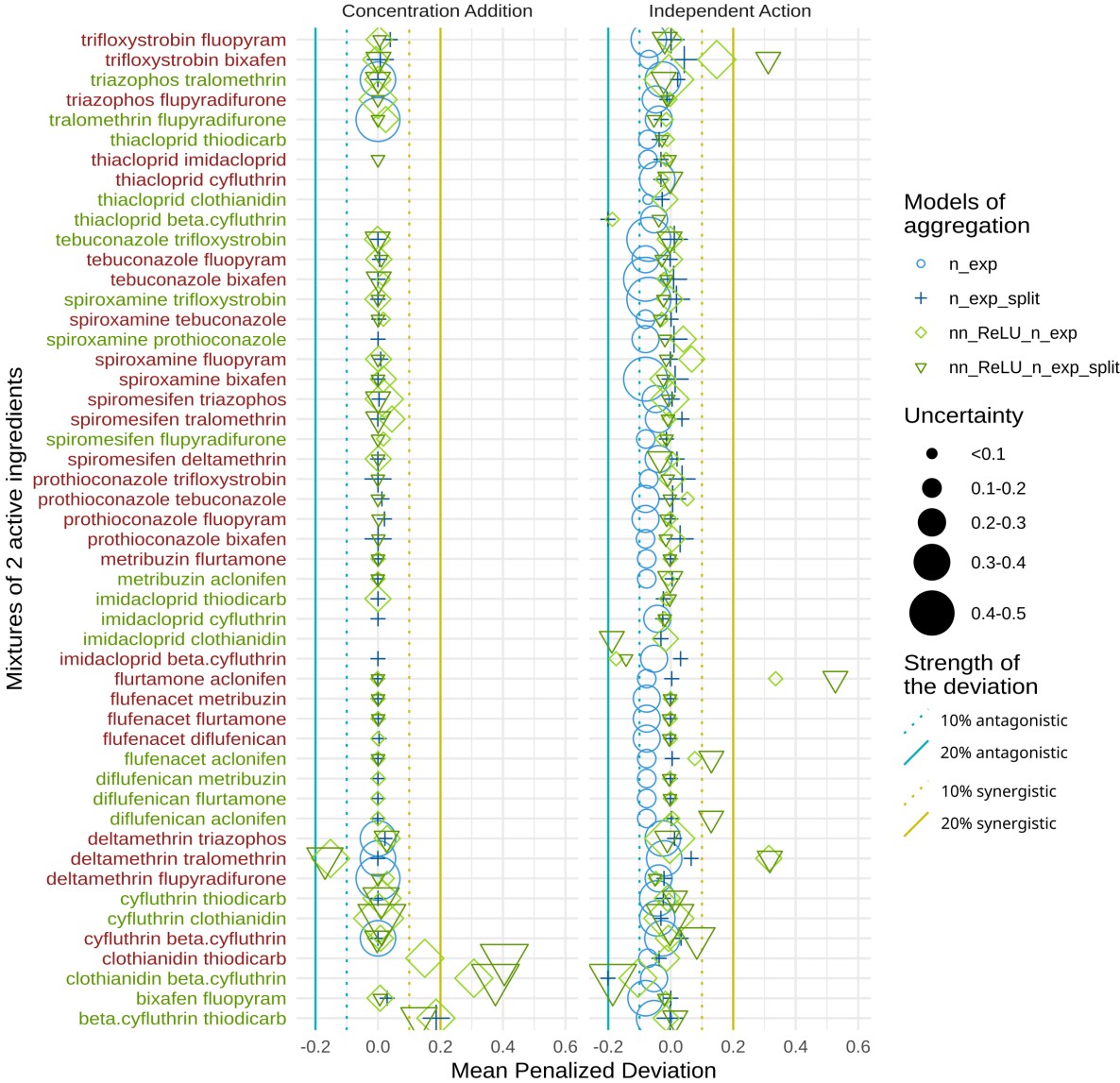

**Fig 7**. **Classification of interaction patterns based on deviations (Mean Peanalized Deviation, MPD, see Eq (11)) from the specified interaction hypotheses, either Concentration Addition (left panel) or Independent Action (right panel), for each mixture combining two active ingredients.** Labels of the y-axis are red when no data exists in the dataset for these mixtures and green when some data points exists in our database. Missing points are due to the 95% uncertainty interval spanning more than 50% of the survival probability range.

herbicides, fungicides, and insecticides. Secondly, our goal was to evaluate the predictive performance of the model for new mixtures, including those with varied active ingredient ratios and entirely novel mixtures. By conducting extensive calibration-validation exercises across different subsampling methods, we observed that incorporating NN elements within the TKTD framework improved calibration quality, but did not consistently enhance predictive accuracy for untested mixtures. Finally, we quantified potential synergistic or antagonistic interactions within these mixtures by assessing deviations from the Concentration Addition (CA) and Independent Action (IA) hypotheses. Our results showed that most of the compound combinations adhered to either the CA or the IA assumptions, in agreement with previous findings [5,8,28].

PLOS Computational Biology

A central finding of this study is a more nuanced understanding of the respective roles of linear and neural network-based models in mixture toxicology. Our results challenge the notion that a more complex NN model should be the default choice. Instead, we propose that the simpler, linear-bridge model should be considered as the parsimonious and interpretable baseline. This model allows for sufficient and more robust predictions for effects of novel mixtures that adhere to additivity assumption, which is critical for regulatory application. The NN-based model is best framed as a powerful, data-intensive exploratory tool, for example when prior toxicological knowledge suggests non-linear interactions, or when model selection criteria (e.g., WAIC or NRMSE) provide strong quantitative evidence that the additional complexity is warranted by the data.

## 4.1 Mixture toxicity with neural-TKTD

Modeling the aggregation of active ingredients from standardized toxicity tests presents substantial challenges due to limited data relative to the vast number of possible mixtures. To address these challenges, we developed a hybrid framework that combines mechanistic (ODE) and non-mechanistic (NN) modeling approaches, leveraging the strengths of each methodology [12,27] to enhance the accuracy and robustness of the predictions. Recent work in pharmacology (PKPD modeling) has demonstrated that integrating pharmacological principles into NN architectures (Neural-PKPD) enables simulations of responses to previously untested dosing regimens [27].

For cross-validation, we tested different data partitioning strategies by splitting data sets into calibration (training) and validation subsets. We found that selecting the entire time series for validation, rather than randomly selecting individual time points, provided the best results. Despite this, model performance across validation subsets was largely equivalent. Calibration results suggest that NN exhibit a degree of overfitting, likely due to their high flexibility, which enables close fits to training data, but may capture noise rather than true underlying distributions. Our initial priors were likely too uninformative, but using Bayesian inference is often subject to criticism for relying on overly informative priors, which can unduly influence the model. For this model study, we did not wish to raise that debate.

A promising outcome of this study is the straightforward protocol developed to build the model (Fig 2) across data sets with varying mixtures and variable exposure profiles, using the number of active ingredients to set layer sizes and employing the *ReLU* activation function. Although the NN architecture is relatively simple by modern standards, it effectively meets our objectives. Using WAIC as a goodness-of-fit criterion, we found that more complex NN architectures (with additional layers and parameters) were less effective in capturing experimental data patterns.

Increasingly, pharmacological studies use hybrid models to capture processes lacking explicit equations. For instance, [22] and [20] implemented NN upstream of pharmacokinetics (PK) to incorporate molecular and mixture-specific parameters as PK inputs, enhancing drug candidate discovery. Given the parallels between TK and PK, future work will extend this approach within the TKTD framework.

## 4.2 Inference of model parameters

For parameter estimation, we employed Bayesian inference, which incorporates prior knowledge and propagates uncertainties [4], allowing for straightforward parameter updates as new data become available. Bayesian inference has recently been used to discern the interactions between covariates and PK parameters, while simultaneously assessing various levels of random effects and measuring uncertainty [16].

In artificial data sets, hybrid models with a single NN layer outperformed traditional linear models in detecting potential synergistic and antagonistic effects. As anticipated, linear models excelled in cases of additive aggregation, as they require fewer parameters for similar fit quality. However, in real data sets, hybrid models enhanced with NN provided superior calibration outputs compared to linear models (Fig 3), reflecting the well-known flexibility of NN to fit diverse functions.

The enhanced performance of NN on real data sets implies that certain compound mixtures do not follow simple additive mechanisms. Calibration-validation procedures identified non-additive interactions for specific compound pairs, with Fig 5 showing non-linear effects for some couples. However, note that in some of these combinations the tested concentration range did not allow strong conclusions, hence, a dataset with more pronounced interaction effects is needed to test the full potential of the model.

### 4.3 Predictability

An additional question concerned whether the effects of new mixtures could be predicted from the marginal toxicities of the active ingredients. TKTD models, with or without neural networks, enable simulation of scenarios lacking empirical data, including varied exposure profiles, varying time points of assessments, assessing different active ingredient ratios, and potentially entirely new active ingredient combinations if they behave similarly to already tested compounds. However, particularly the latter assumption needs to be tested on empirical data.

Previous models for evaluating synergism or antagonism in binary mixtures have relied on isobologram-based statistical tests [35]. Our TKTD approach provides an advantage by enabling predictions across time points through interpolation or extrapolation, offering insights into mixture behaviour over time rather than being limited to specific experimental time points. This dynamic capability improves predictive power by capturing time-dependent interaction patterns.

Using Bayesian inference for calibration allowed uncertainty propagation throughout the predictions. This approach effectively manages uncertainty, yielding low-uncertainty predictions where data support is robust, and avoids unrealistic predictions, such as increased survival at lethal concentrations. This can inform the cost-effective experiment design by prioritizing experiments with high information value given the significant costs associated with experimental toxicology.

Also, a key limitation of the present study is that our model validation was exclusively performed using data from standard acute toxicity tests with constant exposure concentrations only. While this aligns with the calibration dataset, environmental exposures are typically time-variable, occurring as dynamic pulses over time. Therefore, a high-priority for future research is to validate the predictive capabilities of our modelling approach against datasets featuring such time-variable exposure profiles.

### 4.4 Deviation from additivity hypotheses

To test the concentration addition (CA) and independent action (IA) hypotheses, we calculated deviations of the model predictions from expectations under these hypotheses (Fig 7). Deviations from these hypotheses indicate a potential for non-linear aggregation of active ingredients, which is critical for targeted risk assessment. The results suggest that the additivity hypotheses are generally valid, with synergistic effects appearing in the rare instances where deviations are observed. Antagonistic effects were absent, in line with previous studies on interactions of non-additive compounds, where synergistic effects predominate [8,28]. There is no mechanistic foundation that antagonism is less common than synergism, nor is it found less frequently than synergy when reviewing published data [9]. This is despite there being a bias in the literature of looking for synergistic interactions as synergistic mechanisms raise concerns for safety purposes if using a reference model to predict worst case effects, which is not the case for antagonism [28] (e.g., cocktail effects [1]). As a consequence, antagonism is likely to be less scrutinized in scientific studies.

Deviations from the CA hypothesis were observed for two pairs, clothianidin-thiodicarb and clothianidin-beta-cyfluthrin, which would suggest synergistic effects. However, the predictions for both combinations lack consistency (see Fig E in S1 Appendix): e.g., the models predict a decrease in effect with increases in doses in both cases, accompanied by high uncertainty and lack of data for verification. Hence, such results are typically situations of hallucination highlighted by the propagation of uncertainties.

Under the IA hypothesis, strong synergism was observed in the trifloxystrobin-bixafen, deltamethrin-tralomethrin, and flurtamone-aclonifen pairs. However, this is not uncommon and will always occur when concentration-response curves

have steep slopes (>1 for the log-logistic concentration-response model) [9,14] which is the main reason for CA being used as the preferred reference model in a risk assessment context.

The discrepancies that we observed in the deviation between CA and IA models are in line with their underlying assumptions. In the CA model, all substances are summed after estimating the potency, while for the IA model, the responses given in proportions of the control are multiplied. At low concentrations, the mixture of substances below their individual threshold (NEC) still contributes to the mixture effect in CA while fully ignored in IA, while for binary mortality responses using IA, non-lethal effects are ignored. The difference between CA and IA predictions depends on the slope of the dose-response curve. In addition, based on their mathematical definitions, over all concentration ranges, the combined effect in IA tends generally to be lower than what CA would predict for the same concentrations. CA is more conservative compared to IA for safety purposes. Also, the detection of a synergy under IA (a stronger effect observed in data compared to the model prediction) can still be additivity (or even antagonism) under CA [1,14].

We also observed a greater variability in deviation in the IA model compared to the CA models. This is largely because the IA model combines individual responses multiplicatively (or via a function that is non-linear) whereas concentration addition is a linear summation. As a consequence, any variability or uncertainty in the estimation of each individual effect (e.g. due to experimental error, biological variability, or model uncertainty) are much more amplified under IA compared to CA.

These results, which illustrate a low deviation from the additive model, are due to the fact that 90% of the data sets validates the additive hypothesis [8]. Thus, from the present study, if we have shown that theoretically, from artificial data sets, we can retrieve their synergistic or antagonistic characteristics, we demonstrate that the method does not create false positives.

### 4.5 Conclusion

Our framework presents a robust and accessible solution for researchers and practitioners in ecotoxicology, providing a powerful tool for the study and prediction of toxicity results in complex mixtures. By integrating the mechanistic components of TKTD with neural networks within a Bayesian framework, this model improves our ability to understand and quantify interactions among active ingredients, providing information on potential synergistic or antagonistic effects with greater confidence. Bayesian inference enables rigorous parameter estimation and effective uncertainty propagation, essential for reliable predictions and data-informed decision-making. Designed for usability, this advanced framework is both adaptable and practical, promoting its widespread adoption for applications in the evaluation and management of ecotoxicological risks.

In conclusion, while our hybrid model has proven to be a powerful and promising research tool, by demonstrating its utility on standard toxicological data, its transition to a framework ready for widespread regulatory application is contingent upon future validation against datasets featuring time-variable exposures and more pronounced non-additive interactions. We identify this as the critical next step for this research.

### Supporting information

**S1 Appendix. A detailed descriptions of the datasets, a summary of the OECD Test Guideline 203, results from the analysis assessing the potential effects of co-formulants, all cross-validation goodness-of-fit results, and predictive figures.**
(PDF)

**S1 Script. Archive containing all raw and processed data, analytical scripts, and figure-generation code for the study.**
(ZIP)

**S1 Fig. Graphical abstract.**
(TIFF)

## Author contributions

**Conceptualization:** Virgile Baudrot, Sandrine Charles.

**Data curation:** Virgile Baudrot, Thomas Kleiber, André Gergs.

**Formal analysis:** Virgile Baudrot, Sandrine Charles.

**Funding acquisition:** André Gergs, Sandrine Charles.

**Methodology:** Virgile Baudrot, Sandrine Charles.

**Project administration:** André Gergs, Sandrine Charles.

**Software:** Virgile Baudrot, Thomas Kleiber.

**Supervision:** André Gergs, Sandrine Charles.

**Validation:** Nina Cedergreen, André Gergs.

**Visualization:** Virgile Baudrot.

**Writing – original draft:** Virgile Baudrot, Sandrine Charles.

**Writing – review & editing:** Virgile Baudrot, Nina Cedergreen, André Gergs, Sandrine Charles.

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
