## [Decision Letter · Decision Letter 0]

24 Jun 2025

PCOMPBIOL-D-25-00616

A Bayesian Neural Ordinary Differential Equations Framework to Study the Effects of Chemical Mixtures on Survival

PLOS Computational Biology

Dear Dr. Baudrot,

Thank you for submitting your manuscript to PLOS Computational Biology. After careful consideration, we feel that it has merit but does not fully meet PLOS Computational Biology's publication criteria as it currently stands. Therefore, we invite you to submit a revised version of the manuscript that addresses the points raised during the review process.

Please submit your revised manuscript within 60 days Aug 24 2025 11:59PM. If you will need more time than this to complete your revisions, please reply to this message or contact the journal office at ploscompbiol@plos.org. Please include the following items when submitting your revised manuscript:

We look forward to receiving your revised manuscript.

Kind regards,

Samuel V. Scarpino

Academic Editor

PLOS Computational Biology

Mark Alber

Section Editor

PLOS Computational Biology

**Additional Editor Comments:**

I agree with the reviewers that this is an interesting study that is likely to be of relevance in the field. However, I also agree with the reviewers (especially R2), that the comparison between the neural network model and other approaches is difficult to interpret and should be strengthened during revision. Reviewer 2 provides a number of suggestions here, which I suggest the authors pay careful attention to during their revision. I want to stress the importance of using an entirely held-out validation data set for a final comparison. While cross-validation is critical for training, it can be insufficient to rely solely on cross-validation for final model comparison. Additionally, the reviewers noted that more information is needed on how the model was calibrated and how the authors determined that it had fully converged. Lastly, the reviewers raised concerns about whether all necessary data/code/etc were made available. If there are aspects of the study which cannot be released publicly, then the authors must request a waiver per the journal's policy.

**Journal Requirements:**

At this stage, the following Authors/Authors require contributions: Virgile Baudrot, Nina Cedergreen, Thomas Kleiber, Andre Gergs, and Sandrine CHARLES. Please ensure that the full contributions of each author are acknowledged in the "Add/Edit/Remove Authors" section of our submission form.

5) We have noticed that you have uploaded Supporting Information files, but you have not included a list of legends. Please add a full list of legends for your Supporting Information files after the references list.

6) Please ensure that the funders and grant numbers match between the Financial Disclosure field and the Funding Information tab in your submission form. Note that the funders must be provided in the same order in both places as well.

**Reviewers' comments:**

Reviewer's Responses to Questions

**Comments to the Authors:**

Reviewer #1: This manuscript presents the use of an hybrid TKTD (here GUTS) and NN model to study the effects of mixture on the survival of test organisms. I found the manuscript very well written and highly relevant for the field of mechanistic modelling, ecotoxicology, and environmental risk assessment. I would recommend the manuscript for publication provided minor revisions.

Section 2.2.3 The authors used the individual tolerance version of the GUTS model. I would appreciate a reasoning on why this was preferred to the stochastic death one. And if the stochastic death model could be used, what would be the expected differences in performance and predictability?

Section 2.6. Please explain how the calibration was performed and which criteria were used to assess convergence?

Section 2.9. Were the predictions performed with constant exposure only or also with peak or with e.g. FOCUS time-windows? Please precise. If not already performed, some example of predictions with variable exposure would also be appreciated.

Fig 4. Please explain in the caption or in the text the scales used on this figure and their interpretation. e.g. How different are NRMSE values of 0.006 and 0.011 or WAIC values of ca 0 and 3 (panel dataset:add)?

Section 3.3.1 "this deviation metric ranges from 0 to 1". Wouldn't it be from -1 to 1 here?

Discussion:

All of the used datasets used constant concentrations over time. As variable and pulse exposures are more realistic and relevant for risk assessment, could the authors indicate if this was considered and discuss what would be the expected effects on the performance and predictability of the model?

A discussion point on recommendations for practical use, maybe in perspective with the current approaches would be appreciated.

L583 - 587 and 588 - 595 seem to be a repeat/rephrasing of each others. Please check.

Conclusion:

I would appreciate a point in the conclusion on how "ready" do the authors see their model for practical use and which next steps are identified for further research.

Reviewer #2: The manuscript presents a Bayesian neural‐ODE extension of the GUTS TKTD framework to model survival of fish exposed to pesticide mixtures. Integrating a mechanistic TK/TD core with a neural bridge is, in principle, timely and could move mixture toxicology beyond purely additive assumptions. However, several aspects of study design, model justification, transparency, and presentation currently limit the manuscript’s impact and reproducibility.

1. The paper states that “all scripts and data are available in the Supplementary Material” yet the empirical study reports can only be obtained via e-mail request to Bayer. Public repositories (e.g. Zenodo, GitHub) with open licences should host all raw survival counts, exposure concentrations, and analysis code to enable independent reproduction.

2. The authors conclude that “most compound combinations adhered to CA or IA; the added value of neural networks is therefore low” (lines 483-491). Yet the manuscript still promotes the NN as the default choice.

3. Provide a quantitative comparison (e.g. ΔWAIC, NRMSE) between the best linear model and the best NN on an external test set, not only on cross-validation folds. If predictive gains are negligible, a simpler linear bridge may be preferable for regulatory use.

4. Explain why additional regularization (e.g. weight decay, Bayesian hierarchical shrinkage) or simpler priors were not applied given the over-fitting noted by the authors themselves (lines 506-510).

5. Figure 6 shows that when entire binary mixtures are removed from training, linear models often predict better than NN models (lines 405-409). This undermines the claim that the NN generalizes to novel mixtures.

6. Suggest reporting separate performance metrics for:

(a) new time points for known mixtures,

(b) new concentration ratios, and

(c) new ingredient combinations—highlighting where the model does or does not extrapolate.

7.Priors, sampling settings, and convergence diagnostics are not reported. Without them it is impossible to judge parameter identifiability, especially for deeper NNs with many weights. Include a full prior table, sampler settings, and trace/diagnostic summaries in the supplement.

8.Several paragraphs in Sections 4.4–4.5 repeat sentences almost verbatim (e.g. lines 581-595). Please streamline to improve readability.

Minor comments:

1. Clarify the use of “damage addition” versus “concentration addition” and ensure consistent notation in equations (sub- and superscripts).

2. Figures 4-7 colour bars: Provide perceptually uniform colour maps and add units to axes (e.g. concentration in μg L⁻¹).

3. In the figure, the authors express "Figure" but use "Fig." in the main text.

4. The discussion cites recent PK/PD neural-ODE work, but could also discuss contemporary ecotoxicological mixture-omics approaches that leverage self-supervised learning.

Reviewer #3: The manuscript “A Bayesian Neural Ordinary Differential Equations Framework to Study the Effects of Chemical Mixtures on Survival” introduce a new methodology that integrates TKTD models using Ordinary Differential Equations (ODE) with neural networks (NN), offering a robust framework for modeling complex substance interactions. Then this hybrid model was evaluated across 99 acute toxicity studies that included various PPP mixtures, testing its ability to identify and forecast deviations from expected mixture behaviors. This research highlight the potential of combining mechanistic models with machine learning techniques to advance predictive accuracy

in environmental toxicology. Generally, I think this an interesting work with novel method in computational toxicology.

Several suggestions

1. More details on the experiments settings should be provided, at least in SI.

2. For the Fig. 7, what does the x axis mean?

3. GUTS-SD equation should be presented in the main text.

4. As there are many parameters (except NN model) in the model, it is necessary to state clearly how these parameters are estimated. I suggest these estimated parameters could be listed in a supplementary table.

Reviewer #4: This study presents a novel modeling framework for predicting mixture toxicity, integrating neural networks with toxicokinetic–toxicodynamic modeling. The manuscript is well-structured, clearly written, methodologically sound, and provides valuable insights into both model development and toxicological interpretation. The graphical presentations are generally effective, and the discussion is thorough and informative. In my view, the manuscript is almost publishable as is, but I offer the following comments for the authors to consider in their revision for better clarity.

Figure 4: Consider replacing point shapes with circled numbers (e.g., ①②③) to make model distinctions more readable. However, this is a non-essential stylistic suggestion.

Figure 5: The caption claims that point shape indicates cross-validation type (time-points vs time-series), but the legend uses shape for model type. Check to ensure consistency.

Equation 14: Use roman font for “WAIC”.

Line 407, Figure 6: Should “bar(s)” be “points”? Figure 6 uses points, not bars.

Figure 7: The x-axis title is missing.

Figure 8: (a) Is the “Mean Penalized Deviation” here the same as “deviation” in Equation 11? Use consistent terminology to avoid possible confusion. (b) If no observed data exist for red-font mixtures, how was the deviation (as defined in Eq. 11) computed? (c) I suggest revising “Missing point appear when...” to “Missing points are due to the 95% credible interval spanning more than 50% of the survival probability range.”

Lines 448-450: While excluding high-uncertainty data is acceptable for interpretation, showing those data could highlight important research gaps and possible synergistic effects.

Lines 451-452: The sentence “...each line on the graphs represents one of the pairs tested” is unclear. What does the “line” refer to? Horizontal or vertical lines? What each “pair” represents? Chemical pairs? Model-deviation pairs?

**Have the authors made all data and (if applicable) computational code underlying the findings in their manuscript fully available?**

Reviewer #1: Yes

Reviewer #2: **No: **The empirical fish-survival data are proprietary and only available on request from Bayer. This conflicts with open‐science best practices and PLOS policy. All raw survival counts, exposure profiles, and analysis scripts should be deposited in a public repository (e.g., GitHub / Zenodo)

Reviewer #3: Yes

Reviewer #4: Yes

PLOS authors have the option to publish the peer review history of their article (what does this mean?). If published, this will include your full peer review and any attached files.

Reviewer #1: No

Reviewer #2: **Yes: **Wei-Chun Chou

Reviewer #3: No

Reviewer #4: **Yes: **Tan, Qiao-Guo

**Figure resubmission:**
---

## [Decision Letter · Decision Letter 1]

21 Oct 2025

PCOMPBIOL-D-25-00616R1

A Bayesian Neural Ordinary Differential Equations Framework to Study the Effects of Chemical Mixtures on Survival

PLOS Computational Biology

Dear Dr. Baudrot,

Thank you for submitting your manuscript to PLOS Computational Biology. After careful consideration, we feel that it has merit but does not fully meet PLOS Computational Biology's publication criteria as it currently stands. Therefore, we invite you to submit a revised version of the manuscript that addresses the points raised during the review process.

Please submit your revised manuscript within 30 days Dec 21 2025 11:59PM. If you will need more time than this to complete your revisions, please reply to this message or contact the journal office at ploscompbiol@plos.org. Please include the following items when submitting your revised manuscript:

We look forward to receiving your revised manuscript.

Kind regards,

Samuel V. Scarpino

Academic Editor

PLOS Computational Biology

Mark Alber

Section Editor

PLOS Computational Biology

**Journal Requirements:**

At this stage, the following Authors/Authors require contributions: Virgile Baudrot, Nina Cedergreen, Thomas Kleiber, Andre Gergs, and Sandrine CHARLES. Please ensure that the full contributions of each author are acknowledged in the "Add/Edit/Remove Authors" section of our submission form.

5) We have noticed that you have uploaded Supporting Information files, but you have not included a list of legends. Please add a full list of legends for your Supporting Information files after the references list.

6) Please ensure that the funders and grant numbers match between the Financial Disclosure field and the Funding Information tab in your submission form. Note that the funders must be provided in the same order in both places as well.

**Reviewers' comments:**

Reviewer's Responses to Questions

**Comments to the Authors:**

Reviewer #2: I appreciate the authors’ comprehensive and thoughtful revisions. The manuscript is now much clearer, and the framing of the linear versus neural models is balanced and responsible. Most of my original comments have been fully addressed. I have only two minor suggestions for further clarity:

1. Figures 5–6 nicely illustrate the model performance, but it would be helpful to summarize key quantitative metrics (for example, ΔWAIC or NRMSE values) comparing the linear and NN models in a concise table or sentence in the Results section. This would make the relative performance difference immediately visible to readers.

2. Please briefly clarify why new concentration-ratio predictions cannot be separated from new time-point predictions in your validation framework, so that readers understand this methodological constraint.

With these small clarifications, I consider the manuscript fully responsive to prior comments and suitable for publication.

Reviewer #3: My comments are well addressed. The paper can be accepted in the present form.

**Have the authors made all data and (if applicable) computational code underlying the findings in their manuscript fully available?**

Reviewer #2: Yes

Reviewer #3: Yes

PLOS authors have the option to publish the peer review history of their article (what does this mean?). If published, this will include your full peer review and any attached files.

Reviewer #2: No

Reviewer #3: **Yes: **Jianfeng Feng

**Figure resubmission:**
---

## [Editor Report · Decision Letter 2]

30 Oct 2025

Dear Dr Baudrot,

We are pleased to inform you that your manuscript 'A Bayesian Neural Ordinary Differential Equations Framework to Study the Effects of Chemical Mixtures on Survival' has been provisionally accepted for publication in PLOS Computational Biology.

Best regards,

Samuel V. Scarpino

Academic Editor

PLOS Computational Biology

Mark Alber

Section Editor

PLOS Computational Biology

---

## [Editor Report · Acceptance letter]

PCOMPBIOL-D-25-00616R2

A Bayesian Neural Ordinary Differential Equations Framework to Study the Effects of Chemical Mixtures on Survival

Dear Dr Baudrot,

I am pleased to inform you that your manuscript has been formally accepted for publication in PLOS Computational Biology. Your manuscript is now with our production department and you will be notified of the publication date in due course.

With kind regards,

Anita Estes
